# Working Towards Eye Health Equity for Indigenous Australians with Diabetes

**DOI:** 10.3390/ijerph16245060

**Published:** 2019-12-12

**Authors:** Jose J. Estevez, Natasha J. Howard, Jamie E. Craig, Alex Brown

**Affiliations:** 1Wardliparingga Aboriginal Health Equity Theme, South Australia Health and Medical Research Institute, Adelaide SA 5001, Australia; natasha.howard@sahmri.com (N.J.H.); alex.brown@sahmri.com (A.B.); 2Flinders Centre for Ophthalmology, Eye and Vision Research, Department of Ophthalmology, Flinders University, Adelaide SA 5042, Australia; jamie.craig@flinders.edu.au; 3Faculty of Health and Medical Sciences, University of Adelaide, Adelaide SA 5005, Australia

**Keywords:** Indigenous Australians, blindness, diabetic retinopathy, vision impairment, eye health, equity, inequality, vision loss, diabetes

## Abstract

Type 2 diabetes mellitus (T2DM) poses significant challenges to individuals and broader society, much of which is borne by disadvantaged and marginalised population groups including Indigenous people. The increasing prevalence of T2DM among Indigenous people has meant that rates of diabetes-related complications such as blindness from end-stage diabetic retinopathy (DR) continue to be important health concerns. Australia, a high-income and resource-rich country, continues to struggle to adequately respond to the health needs of its Indigenous people living with T2DM. Trends among Indigenous Australians highlight that the prevalence of DR has almost doubled over two decades, and the prevalence of diabetes-related vision impairment is consistently reported to be higher among Indigenous Australians (5.2%–26.5%) compared to non-Indigenous Australians (1.7%). While Australia has collated reliable estimates of the eye health burden owing to T2DM in its Indigenous population, there is fragmentation of existing data and limited knowledge on the underlying risk factors. Taking a systems approach that investigates the social, environmental, clinical, biological and genetic risk factors, and—importantly—integrates these data, may give valuable insights into the most important determinants contributing to the development of diabetes-related blindness. This knowledge is a crucial initial step to reducing the human and societal impacts of blindness on Indigenous Australians, other priority populations and society at large.

## 1. Introduction

Type 2 diabetes mellitus (T2DM) represents a global health epidemic of critical importance that poses significant public health challenges to current and future generations [1]. Recent estimates from the International Diabetes Federation suggest that 451 million people globally had T2DM in 2017 and that number is projected to increase, reaching 693 million by 2045 [2]. Even these estimates are likely to be an underestimation owing to missing data from several countries [3]. Nevertheless, it is well recognised that much of the burden of T2DM is borne by disadvantaged and marginalised groups, including Indigenous people [4]. Worldwide, Indigenous people represent a population of at least 370 million, approximately 5% of the world’s population, including 5000 distinct cultures from over 90 different countries [5]. Despite the rich heterogeneity in culture, geographic location, history and language, increasing and inequitable rates of T2DM in comparison to non-Indigenous populations is a shared phenomenon, particularly among Indigenous people within high-income countries [6]. Additionally, the rates of diabetes-related complications among Indigenous people, such as macrovascular disease, retinopathy and end-stage renal disease, are disproportionately higher than their non-Indigenous counterparts, representing a major cause of morbidity and mortality [6]. 

Australia, a high-income and resource-rich country, continues to struggle to adequately respond to the health needs of Aboriginal and Torres Strait Islander people (hereafter respectfully referred to as ‘Indigenous Australians’), particularly those suffering from or at risk of T2DM [7,8]. For the Indigenous people of Australia, the ongoing impact of colonisation and continuing social and political oppression and dispossession have contributed to significant socio-economic and health inequities across several key health indicators [9]. T2DM is one of the most important contributors to health inequities (both mortality and morbidity) between Indigenous and non-Indigenous Australians, as it is a potent driver of premature onset cardiovascular, renal and retinal complications [10,11,12].

## 2. Diabetic Retinopathy among Indigenous Australians

The increasing prevalence of T2DM has meant that vascular complications leading to vision impairment and eventual blindness, specifically diabetic retinopathy (DR) and its vision-threatening endophenotypes, diabetic macular oedema and proliferative diabetic retinopathy, have persisted as key population health concerns among Indigenous Australians [13,14]. Secular trends have demonstrated an almost doubling in the prevalence of DR in the last twenty years. The earliest reports from 1996, albeit from a specific region of Australia, found that 20.9% of the population had any DR. By comparison, the most recent national level estimates were 39.4% in 2017 among Indigenous Australians over the age of 40 [14,15,16]. Furthermore, the prevalence of diabetes-related vision impairment, although varying widely between 5.2% to as high as 26.5% depending on the study context, sampling methodology and region, is consistently reported to be higher among Indigenous Australians [14,15,16]. Among non-Indigenous Australians, diabetic complications contribute only 1.7% of the total burden of vision impairment and these rates have remained largely unchanged over a twenty-year timespan [12,17]. The increasing availability of highly effective but expensive sight-saving treatments (i.e., intravitreal anti-vascular endothelial growth factor or corticosteroid drugs), improved overall diabetes and risk factor control, and access to a range of DR screening programs are likely to have played a role in this stability [18]. Yet, Indigenous Australians have not experienced the same degree of success, nor have they experienced the same eye health outcomes. 

Individuals with T2DM are at an additional heightened risk of developing a secondary or coexisting ocular condition. In fact, Indigenous Australians with T2DM seem to be at an increased risk (up to 8.5-fold) of vision impairment from non-DR-related causes, likely through developing earlier-onset cataract and experiencing refractive error changes [19]. Given that more than 30% of Indigenous Australian adults have T2DM, and that T2DM is increasingly being diagnosed in children and young adults, it is anticipated that vision impairment from non-DR-related causes could also rise [14,20]. Even with a significant body of research into the burden of T2DM and DR being available from other countries, existing services and approaches in Australia seem incapable of reversing these emerging trends.

## 3. Progressing Towards Eye Health Equity

Granting that there are existing health system insufficiencies concerning the delivery, accessibility, coordination and cultural appropriateness of eye health services, some notable and positive progress has recently been achieved. Firstly, in the most recent national epidemiological survey of eye health in Australia, the rates of diabetes-related vision impairment were 5.2% among Indigenous Australians, approximately three-fold the prevalence seen in non-Indigenous Australians [15]. However, whilst inequity between Indigenous and non-Indigenous Australians still exists, the crude rates appear to have fallen from 12% as identified in a similar survey conducted in 2008 [21]. Secondly, although the prevalence of ‘any’ DR is relatively high (39% among individuals with self-reported diabetes), three-quarters of these cases were considered low-grade and non-sight-threatening (minimal or mild non-proliferative DR) [15]. Finally, the number of Indigenous Australians who had received their yearly DR screening examination, as recommended by national clinical guidelines [22], has increased from 20% to 53% over 10 years, and this speaks to improvements in healthcare access [15,23].

National government-funded initiatives have also been implemented, aimed at reducing the eye health disparities among Indigenous Australians living with T2DM. These include: (1) a new Medicare Benefits Schedule (a list of the medical services for which the Australian Government will pay a Medicare rebate, to provide patients with financial assistance towards the costs of their medical services) item enabling the billing of DR detection services using a non-mydriatic retinal camera by non-ophthalmic clinicians (such as health workers, nurses, primary care physicians and endocrinologists); (2) additional equipment and training for health workers involved in the delivery of Aboriginal and Torres Strait Islander eye health services; (3) funding for continuing and increased specialist visits by optometrist and ophthalmologists delivering care in Indigenous Australian communities; and (4) investments in novel models of eye care delivery such as telehealth, artificial intelligence, regional eye health hubs and mobile DR screening units. Tracking of current initiatives and monitoring trends at a national level, disaggregated by important social measures such as gender, geography and socioeconomic status, must also underpin this work so that the many stakeholders involved can monitor—and remain accountable for—meaningful progress. Further substantial investments are also needed to sustain, strengthen and evaluate eye health programs, as it has long been appreciated that almost all cases of severe vision loss arising from DR are unnecessary and preventable with early detection and timely treatment [24,25]. Nonetheless, active partnerships between Indigenous Australian people, communities and organisations are fundamental in improving the eye health outcomes of Indigenous Australians.

## 4. Addressing the Knowledge Gaps

Even though numerous risk factors have been described in the literature influencing the rapid and unprecedented rise in T2DM prevalence rates in Indigenous populations around the globe [26], fragmented and insufficient data are available on the factors contributing to many of the downstream complications such as retinopathy [6]. For example, large population-based studies to date have only provided evidence for the epidemiology of DR alongside investigations into a narrow range of risk factors, which have identified geography (remote, OR 2.46), diabetes duration (OR 1.69), and a glycosylated haemoglobin percentage (HbA1c) above 7% as being associated with having DR among Indigenous Australians [15,27]. These studies have not investigated the complex T2DM/DR phenotype and the underlying social, psychological, environmental, behavioural, clinical, biological, metabolomic and genetic risk factors. The available data to date have instead focused primarily on “surveying” vision impairment rates and causes, and eye health outcomes alone, without investigating the broader determinants that drive DR-related blindness. Taking a systems approach, which involves the deep characterisation of participants and synergy of data across the clinical, social, biological and genetic platforms, with as many environmental factors as possible [28], could unravel the mechanisms that are most influential on diabetes-related blindness. Future population-based eye studies including Indigenous Australian people in their sample should consider integration of whole systems and prospective analyses into their study methodology, with detailed characterisation of not only eye complications, but sociodemographic, clinical and biological measures.

The risk factors of interest to sight-threatening DR may be fundamentally psychosocial, environmental, clinical, biological or genetic in nature, but are best interpreted together given the complexity of the T2DM phenotype and its vascular complications [18,28]. Non-modifiable and modifiable risk factors such as living remotely, duration of diabetes, increasing age, access to health care and poor glycaemic control are likely to also play an important role in the development of sight-threatening DR [15,27]. However, health inequities based on gender and socioeconomic standing can further influence the wellbeing of an individual, with reports suggesting that socioeconomic factors alone could explain more than two-thirds of the global variation in prevalence of vision impairment and blindness [29]. Other risk markers such as hypertension, dyslipidaemia, obesogenic traits and inflammatory biomarkers also need to be empirically explored and may well enhance our understanding of what drives the eye health inequities between Indigenous and non-Indigenous Australians and what can be done to overcome them [18,30].

Population-specific genetic investigations will further provide useful insights as DR is a complex disease with likely gene–environment and gene–gene interactions. As an example, the strongest known clinical risk factors for DR, duration of diabetes and glycaemic control (collectively glycaemic exposure), were estimated to contribute only a small proportion of the variation in risk in major clinical trials conducted in the United Kingdom and the United States [31]. Therefore, it is hypothesised that ethnic differences, environmental factors and genetic susceptibility could also play a role in DR progression, but whether genetic factors, or others, are intrinsically involved in the premature and severe vision impairment experienced by Indigenous Australians is as yet unknown [32]. It is important to search for all factors that contribute to DR risk, both biological and non-biological, to identify more complete social, clinical or biological markers that can be targets for intervention to prevent, delay or predict the progression of DR. It is anticipated that greater knowledge on the underlying social, clinical and biological determinants will lead to the development of better treatments, or enhance existing therapies, and enable improvement of clinical management and health system service delivery that: (1) delays the onset of T2DM and related complications; (2) slows the progression of vision loss in those with already established disease; and (3) decreases the rate of blindness-related morbidity.

## 5. Conclusions

T2DM and its complications are pressing global health challenges for the Indigenous people of the world, including Indigenous Australians. Although there have been important eye health investments and initiatives implemented, inequities in eye health outcomes persist between Indigenous and non-Indigenous Australians living with T2DM. Overcoming these inequities requires a systems approach and integration of existing data, with a compelling need to identify the underlying clinical, biological and genetic risk factors that lead to blindness. Addressing these knowledge gaps is paramount to progressing towards eye health equity and may provide better risk stratification, initiation of more targeted therapies or public health programs and a personalised approach to blindness prevention that is relevant for Indigenous Australians with T2DM. Importantly, greater consideration is also needed into addressing complex factors related to the social determinants of health, like access to health care, gender inequities and challenges delivering eye care across diverse geographic regions. Transformational changes across the prevention, treatment and management continuum are required to address the eye health disparities between Indigenous Australians and non-Indigenous Australians, whereby some of the successful initiatives and learnings can be adapted for global priority populations.

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
