# Peer review of "Working Towards Eye Health Equity for Indigenous Australians with Diabetes"

_ijerph, 2019, doi:10.3390/ijerph16245060_

Round 1

Reviewer 1 Report

I consider this Ms to have valuable data that would be of interest of our readers. This is a nice review of the DM problems among Indigenous Australians and I will reccomend a fast publication. 

Author Response

We thank the reviewer for their expertise.

Reviewer 2 Report

This study has been well-highlighted and summarized the issue of eye health equity for indigenous Australian with diabetes. 

Author Response

We thank the reviewer for their expertise and time taken to read our work.

Reviewer 3 Report

I congratulate the authors for this commentary on equitable eye care for Indigenous Australians with diabetes. 

This paper would be improved by clarification of some of the statistics used. Although the authors note that fragmented data is an issue, this data is then presented as if it were national level evidence. The limited availability of data for assessing temporal/national trends should be highlighted, rather than glossed over in places. 

In addition, the authors state in the abstract that although Australia now has reliable estimates of the eye health burden owing  to T2DM in its Indigenous population, there is fragmentation of existing data and limited knowledge on the underlying risk factors. I feel that the paper would be strengthened by more detailing of what specific knowledge of risk factors would be beneficial.

The authors also mention taking a systems approach in several occasions, however haven't fully explained why this may be beneficial, apart from the possibility of identifying/unraveling 'determinants' or 'mechanisms'. I don't feel that the conclusion: "Overcoming these inequities require a systems approach and integration of existing data, with a compelling need to identify the underlying clinical, biological and genetic risk factors that lead to blindness" has been sufficiently justified in the preceding text. 

Some specific comments:

Page 2, lines 15-17:

"Secular trends have demonstrated an almost doubling in the prevalence of DR from 20.9% in 1996 to 39.4% in 2017 among Indigenous Australians over the age of 40"  

Please contextualise the prevalence statistics reported here. It appears as if the authors are comparing a 1996 finding from the Katherine region with the national average from the National Eye Health Survey, and implying this is national secular trend. Please clarify, or provide additional context around the appropriateness of the comparisons. 

Page 2, lines 46-49:

"Finally, the number of Indigenous Australians who had received their yearly DR screening examination, as recommended by national clinical guidelines,[22] has increased from 20% to 53% over 10 years, and this speaks to improvements in healthcare access.[15,21]"

I was unable to find the reported statistics about the reported increase in annual DR screening in the provided references. It is unclear if that improvement in annual DR screening is a national average, or at discrete locations. If the latter, please provide appropriate context.

Page 3 lines 44-45:

"It is important to search for factors that contribute to the other ~90% of risk, both biological and non-biological, to identify more complete targets to prevent, delay or predict the progression of DR"

Please clarify what is meant by 'targets'. Currently does this section imply that 90% of the risk for DR progression is unknown?

Page 3, lines 23-25:

"Taking a systems approach, which involves the synergy of data across the clinical, environmental, social, biological and genetic platforms, could unravel the mechanisms that are most important for diabetes-related blindness.[26]"

It is unclear what is being referenced in this sentence and what are the authors original thoughts. Please ensure that this is clear and obvious throughout, I should not have to guess when concepts are being referenced,  or whether they are original concepts. 

Author Response

Please see the attached word document.
